# Hybrid Ultra-Low-Radioactive Material for Protecting Dark Matter Detector from Background Neutrons

**DOI:** 10.3390/ma14133757

**Published:** 2021-07-05

**Authors:** Marina Zykova, Mikhail Grishechkin, Andrew Khomyakov, Elena Mozhevitina, Roman Avetisov, Nadezda Surikova, Maxim Gromov, Alexander Chepurnov, Ivan Nikulin, Igor Avetissov

**Affiliations:** 1Department of Chemistry and Technology of Crystals, Mendeleev University of Chemical Technology, 125047 Moscow, Russia; zykova_mp@inbox.ru (M.Z.); grimb@mail.ru (M.G.); homa-ifh@yandex.ru (A.K.); spring_storm@mail.ru (E.M.); armoled@mail.ru (R.A.); 2Skobeltsyn Institute of Nuclear Physics, Lomonosov Moscow State University, 119234 Moscow, Russia; ndsurikova@yandex.ru (N.S.); gromov@physics.msu.ru (M.G.); aschepurnov@yandex.ru (A.C.); 3Joint Institute for Nuclear Research, 141980 Dubna, Russia; 4Department of Engineering Material Science, Belgorod State National Research University, 308015 Belgorod, Russia; nikulin@bsu.edu.ru

**Keywords:** polymethylmethacrylate, gadolinium, uranium, thorium, hybrid material, inductively coupled plasma mass spectrometry, low-radioactivity, dark matter, neutron background

## Abstract

A laboratory technology for a new ultra-low background hybrid material (HM) which meets the requirements for neutron absorption with simultaneous neutron detection has been developed. The technology and hybrid material can be useful for future low background underground detectors designed to directly search for dark matter with liquid noble gases. The HM is based on a polymethylmethacrylate (PMMA) polymer matrix in which gadolinium nuclei are homogeneously distributed up to 1.5 wt% concentration in polymer slabs of 5 cm thickness. To determine the 65 impurity elements by the inductively coupled plasma mass-spectrometry (ICP-MS) technique in the Gd-based preparations in 100–0.01 ppb range, the corresponding method has been developed. Limits of determination (LD) of 0.011 ppb for uranium, and 0.016 ppb for thorium were achieved. An analysis of Gd raw materials showed that the lowest contents of U and Th (1.2–0.2 ppb) were detected in commercial Gd-based preparations. They were manufactured either from secondary raw materials (extraction phosphoric acid) or from mineral raw materials formed in sedimentary rocks (phosphogypsum). To produce the Gd-doped HM the commercial GdCl_3_ was purified and used for synthesis of low-background coordination compound, namely, acetylacetonate gadolinium (Gd(acac)_3_) with U/Th contents less than LD. When dissolving Gd(acac)_3_ in methylmethacrylate, the true solution was obtained and its further thermal polymerization allowed fabrication of the Gd-doped PMMA with ultra-low background.

## 1. Introduction

The direct search for dark matter (DM) in the form of weakly interactive massive particles (WIMPs) is one of the quickly developing experimental methods aimed at finding answers to the fundamental problem of the nature of DM in modern physics. The WIMPs could be detected through their elastic scattering on target nuclei. Low-energy (<100 keV) nuclear recoils are produced as a result of this process and they are registered by the experimental setup. In this case, the neutron background is one of the most unpleasant and requires maximum suppression. The interaction of neutrons with a target leads to a nuclear recoil event and, therefore, may mimic the scattering of DM particles. The placement of the detectors in underground laboratories and the use of ultra-low-background materials make it possible to achieve uniquely low background levels. However, even in underground laboratories, it is necessary to take additional measures to protect the sensitive target of a detector against cosmogenic neutrons [1] and neutrons born in the (α, *n*) reactions [2]. An effective method of protection against neutron background is the use of a special ultra-low background hybrid material for absorbing neutrons in the design of the detector. This material should not only efficiently thermalize and absorb the neutrons, but also inform the detector about neutron capture. This functionality could be achieved by the use of a hybrid material consisting of Gd-doped plastic. The plastic effectively slows down neutrons due to high hydrogen concentrations while the gadolinium with natural concentration of ^155,157^Gd isotopes has unique thermal neutron capture efficiency [3,4,5,6,7]. Gamma rays (four gamma-quanta on average) with total energy up to 8 MeV accompanying neutron capture on ^155,157^Gd nuclei could serve as a signature of the event. One could detect and count these signatures by the scintillation technique.

However, the material to be used in DM detectors should also meet the requirement of ultra-low background properties, namely, it should have as little of its own radioactive background as possible with special attention paid to the concentrations of alpha-active isotopes, which are the sources of secondary neutrons due to the (α, *n*) reactions. The most widely spread alpha-active isotopes are isotopes of U and Th. Therefore, the materials from which the detector’s parts are made should have a residual concentration of U and Th isotopes less than 10 and 25 ppt (1 × 10^−9^ and 2.5 × 10^−9^ wt%), respectively, in other words, or less than about 100 μBq/kg.

For instance, low-background materials are necessary to manufacture the parts of future low background underground detectors for direct DM search based on liquid noble gases. These parts are designed to protect the central target from background neutrons. These parts of the detector surrounding the central sensitive volume could be called an external detector (ED), inform about the current neutron background and protect the detector’s target from neutrons. Such an ED performs the functions of thermalization, absorption and accompanying detection of background neutrons. It is dedicated to protecting the main target for dark matter detection from background neutrons and implementing a veto mechanism for events associated with background neutrons as described in Ref. [8].

Note that an ultra-low-background Gd-doped hybrid material for ED is required on an industrial scale. If the size of the central target has a scale of a few cubic meters, the ED needs tens of tons of the material containing at least a few hundred kilograms of the Gd-based component in terms of elemental gadolinium. In this regard, strategically, the development of the technology of an ultra-low-background hybrid Gd-containing material should be carried out in parallel with the development of a technology for the purification of a gadolinium preparation from radioactive impurities and the choice of the material of the organic matrix. These tasks were solved in the framework of this study.

## 2. Theoretical Issue

The number of background neutrons $N_{n}$ emitted as a result of the (α, *n*) reactions in a material during the characteristic operating time of the detector can be calculated by the equation
(1)$$N_{n}=\;\left(1-\varepsilon \right)mT\underset{i}{\sum }A_{i}Y_{i},$$
where ε is the detector efficiency of neutron selection, m is the mass of the material, T is the period of data taking, $A_{i}$ is the total specific activity (per unit mass of the material) for a part or complete decay chain designated by the index i and $Y_{i}$ is the respective neutron yield. For calculations, the characteristic period of the data collection is often chosen to be 10 years. The exposure is defined as the product of the target mass within the fiducial volume and time.

In the current study, it is necessary to minimize the number of neutrons from the (α, *n*) reactions in the new material. Taking into account Equation (1), this goal can be achieved by changing the following parameters: the mass of the material m, the total activities $\left\{A_{i}\right\}$ and the neutron yields $\left\{Y_{i}\right\}$.

Since the expected composition of the hybrid material includes about 95% polymethylmethacrylate (PMMA) by mass and only about 5% of some gadolinium compound, it is impossible to significantly reduce the number of background neutrons by changing the material mass.

The second class of parameters is the set of neutron yields $\left\{Y_{i}\right\}$. Its impact on the final result is not obvious. For example, light isotopes, such as boron or fluorine, which might be a part of the gadolinium compound with the mass fraction of <1 wt% or might be included in the hybrid material in the form of residual technological contaminants, can increase the neutron yields by tens of percent as shown in Table 1.

In the last years, the idea of introducing gadolinium oxide Gd_2_O_3_ into PMMA has become popular due to Gd_2_O_3_ being available on the market, it is relatively pure in uranium and thorium and practically does not change the neutron yields of the final material compared to PMMA without additives. An alternative approach is based on the formation of a real solution of the Gd-based compound in methylmethacrylate (MMA). For instance, it may be a dissolution of a complex compound—gadolinium acetylacetonate (Gd(C_5_H_7_O_2_)_3_) in the initial MMA monomer, followed by polymerization and the production of a hybrid Gd-PMMA material. In terms of its constituent chemical elements, it is similar to PMMA and does not contain isotopes with extremely high neutron yields from the (α, *n*) reactions. The neutron yields for gadolinium oxide and acetylacetonate were calculated using the NeuCBOT [2,9] and TALYS-1.95 [10,11] programs. The radioactive series ^232^Th, ^235^U and ^238^U were taken into account. The last one was divided into three parts: the upper (starts from ^238^U), middle (from ^226^Ra) and lower (from ^210^Pb) chains. It has done this way because the secular equilibrium might be broken for the ^238^U decay chain, but it is expected to be preserved in each sub-chain [9]. Unlike other isotopes in the radioactive series, ^226^Ra and ^210^Pb have long half-lives, 1600 and 22 years, respectively, and can be accumulated in materials due to different processes [9]. The calculation results are shown in Table 1. Analysis of the calculated data showed that the neutron yields are almost the same for pure PMMA, for PMMA with gadolinium oxide in the amount of 1.5 wt% Gd in terms of the mass of gadolinium and for PMMA with gadolinium acetylacetonate with the same mass fraction of 1.5 wt% Gd. Thus, a reduction in the number of background neutrons from the hybrid material can only be achieved by reducing the total activities, or, in other words, by reducing the levels of contamination of the entire hybrid material with uranium and thorium. As mentioned in the introduction, the total specific activity level required for the detector materials should not exceed 100 μBq/kg.

## 3. Materials and Methods

### 3.1. Impurity Determination by ICP-MS

To date, the most readily available Gd-contained chemicals on the market are oxide, chloride, nitrate, and sulfate. Therefore, gadolinium oxide and its salts were selected as the target products for creation of Gd-contained hybrid material with low radioactivity background. To analyze chemical purity of Gd-based preparations we used inductively coupled plasma mass-spectrometry (ICP-MS) with preliminary transfer of a solid sample to liquid phase. For water soluble preparations, namely, gadolinium nitrate (Gd(NO_3_)_3_), gadolinium sulfate (Gd_2_(SO_4_)_3_) and gadolinium chloride (GdCl_3_) we used extra pure water (AquaMax-Ultra 370 Series, Young Lin Instruments Co. Ltd., Anyang, Korea) with a specific resistance of 18 MΩ·cm. Gadolinium oxide (Gd_2_O_3_) was dissolved in high-purity nitric (HNO_3_) acid (7N7) purified by a Berghof BSB-939-IR surface distillation system in a SPEEDWAVE-FOUR microwave decomposition system (BERGHOF GmbH & Co. KG, Germany, Drolshagen) equipped with DAP-100 PTFE autoclaves (BERGHOF GmbH & Co. KG, Germany, Drolshagen).

Polymer materials were dissolved in high-purity sulfuric (H_2_SO_4_) acid (8N Ultrapur, Sigma Aldrich, St. Louis, MS, USA) by using the same microwave decomposition system.

Analytical measurements were carried out on a NexION 300D inductively coupled plasma mass spectrometer (PerkinElmer Inc., Waltham, MA, USA).

We used the TotalQuant method [12] for determination of 65 chemical elements’ concentrations. The quantitative analysis of Th and U was carried out using the “additives” method taking into account the concentration of the main (matrix) elements in the analyzed solution. The standard solutions (PerkinElmer Inc.) were used for calibration. The optimized operating mode of the NexION 300D spectrometer for impurity analysis of samples with Gd matrix element is presented in Table 2.

Natural gadolinium is the mixture of seven stable isotopes: ^152^Gd (0.20%), ^154^Gd (2.18%), ^155^Gd (14.80%), ^156^Gd (20.47%), ^157^Gd (15.65%), ^158^Gd (24.84%) and ^160^Gd (21.86%) [13]. The presence of a large number of isotopes in the matrix element of the analyzed solution inevitably leads to mass spectral overlays (interferences) caused by isobar overlays or polyatomic ion overlays based on matrix elements, solvents and plasma forming gas (Ar) (Table 3). In the case of analysis of Gd-based samples containing rare earth elements, there is another problem associated with the significant influence of matrix elements: signal suppression, the formation of polyatomic ions (MO^+^, MOH^+^) and ions with a double charge (M^++^). These effects result to generation of areas at mass spectra that require detailed analysis.

When an impurity element ion and a polyatomic ion have significant difference in effective radii (for instance the formation of Gd^16^O^+^, Gd^16^O^1^H^+^, Gd^40^Ar^+^), interferences can be eliminated in an ICP-MS collision/reaction cell using the Kinetic Energy Discrimination (KED) mode [14]. The analysis of 192–200 mass spectrum range in the KED and standard modes showed that helium entering the cell collapse more efficiently with large ions, such as Gd^40^Ar^+^ (Figure 1c,d). This led us to identify the Pt isotope with minimal overlay of Ar, which dominated in plasma. The problem was solved by calculation of Pt concentration using the ^194^Pt isotope with a natural abundance of 32.9%. We failed to successfully use the KED mode for 166–177 mass spectrum range, which was associated with the comparatively large scale generation of hydroxide and oxide groups’ ions (Figure 1a,b).

There was an attempt [15] to solve this problem by optimization of the conditions for the formation of ions in Ar plasma at which the Ln^2+^/Ln^+^ level was maximum and the LnO^++^/LnO^+^ level was minimal. In the case of gadolinium, the authors failed to solve the problem because the proportion of the formation of interfering doubly charged hydroxyl ions of the matrix element Gd^16^O^1^H^++^/Gd^16^O^1^H^+^ was more than 0.2%. This did not allow separating chemical elements such as Yb and Lu.

Table 4 presents the chemical elements which caused difficulties in the analysis of Gd-based solutions and the corresponding isotopes of these elements recommended for quantitative determination.

To reduce the limit of determination (LD) of U and Th the “additives” method was used. The standard solutions of uranium (Uranium (U) Pure Plus Standard, 1 μg/mL, PerkinElmer Inc., Boston, MA, USA) and thorium (Thorium (Th) Pure Plus Standard, 1 μg/mL, PerkinElmer Inc., Boston, MA, USA) were used for calibration. We succeeded in reducing the LD for uranium to 0.011 ppb and for thorium to 0.016 ppb for 95% confidence probability.

### 3.2. Analysis of Gd Concentration Distribution

Gd-concentration distribution inside Gd-doped hybrid material was performed by a scanning electron microscope (VEGA-3 LMU, TESCAN Corp., Kohoutovice, Czech Republic) equipped with an energy-dispersive unit (EDS Inca Energy X-MAX-50, Oxford Instruments, Abingdon, UK). AZTec software was used for data gathering and processing. The beam energy was 20 KeV.

### 3.3. Mechanical Test of Polymer and Hybrid Materials

Mechanical tensile tests of PMMA and Gd-doped hybrid material (HM) samples were carried out using an Instron 5882 (Instron, Norwood, MA, USA) testing machine at 25 ± 1 °C (298 K) and at temperature of liquid nitrogen (77 K). The samples were cut from PMMA or HM bulks in the form of 2 mm thickness plates. The plates were fixed in the grips of the testing machine and loaded to 5 N. For low-temperature measurements, the grips with the sample were placed into a container, into which liquid nitrogen was gradually added. Simultaneously the load of the sample was monitored all the time (no more than 10 N). The loading was reduced (if necessary) by bringing the grips together. After the sample was completely immersed in liquid nitrogen, it was maintained for about 3 min in order to eliminate the temperature gradient. The sample was then loaded with an increasing load. The loading rate was 1.0 ± 0.1 mm/min. The greatest strength was recorded at the moment of failure to determine the tensile strength. Samples that failed outside the test section were not included in the result of the experiment. The arithmetic means of at least 10 samples’ tests were taken as the result of measuring the strength, elongation and elastic modulus.

## 4. Results and Discussion

The general problem of the research was dealing with the reduction of U and Th concentrations in Gd-doped organic-inorganic hybrid materials (HM).

As the first stage of production of Gd-doped HMs with low radioactivity background, we need to obtain an inorganic Gd-based preparation with low radioactivity background. There is no fundamental approach, which allows reducing the concentration of certain elements to the ppt-level at the content of other impurities at ppm-level.

There are several probable ways to solve this problem for U and Th: (1) A general purification of Gd-based preparations by extraction techniques, (2) selective extraction of U and Th using specific reagents, (3) conversion of Gd-based preparation in a volatile compound and further distillation or sublimation processing and (4) a multistage approach which could be a combination of all the above techniques.

The choice of the most efficient strategy of purification was made after analysis of the commercial Gd-based preparations, which enabled us to discover what level of purification must be fulfilled.

### 4.1. Analysis of Commercial Gd-Based Preparations

At present, the majority of Gd-based preparations are synthesized from raw materials mined in China. Analysis of two different batches of Gd_2_O_3_ (Yeemeida Technology Co. Ltd., PRC), showed that they differed significantly in uranium concentration (Figure 2). This indicated, with high probability, that these preparations were obtained from different raw source materials.

Generally, for Gd-based preparations the manufacturers do not conduct precise control on the impurities of Th and U, but focus on the residual rare earth elements and the total impurity concentration (Figure 3). The deviations in the impurities’ distribution at an overall low concentration of impurities do not allow relying on reproducibility in the levels of residual radioactive impurities.

It was found out that the Gd(NO_3_)_3_ preparation manufactured by FSUE “IREA”, gadolinium oxide from the same manufacturer, gadolinium oxide produced by LANHIT LLC, and by Shin-ETSU, had the highest chemical purity. A distinctive feature of these manufacturers is their approach to the selection of raw materials, which allows us to obtain preparations of high chemical purity in the future. All analyzed preparations are characterized by a similar content of elements such as sodium, lead and iron. Their concentrations range from 10^−3^ to 10^−5^ wt%. The chemical purity for all analyzed Gd_2_O_3_ preparations, except IoLiTech, was no less than 99.987 wt% for 65 impurity elements (Figure 3). The composition of impurities for the analyzed preparations varied significantly in the area of heavy elements. The most impurities are in the region of rare-earth (RE) elements, due to their proximity in terms of physicochemical properties to gadolinium.

Analysis of the Gd-based commercial preparations showed that preparations obtained from secondary raw materials (extraction phosphoric acid—EPA) or from mineral raw materials formed in sedimentary rocks (phosphogypsum) were the best suited to produce a preparation with extremely low radioactive background. For further purification we gave preference to soluble salts rather than gadolinium oxide. The last argument is related to the fact that most of the known methods for purification of preparations containing REs are based on solution technologies, or on technologies for the separation of volatile components (most often chlorides) under reduced pressure.

### 4.2. Purification of GdCl_3_ Preparations

We chose GdCl_3_ commercial preparation for further purification. There were two reasons: (1) the raw source was enough to supply more than one ton and (2) GdCl_3_ has high vapor pressure and is suitable for vacuum sublimation. Thus, we can develop the purification process for the known impurity distribution for this preparation and be sure that the total amount of the raw material will be enough to solve the general task. We decided to develop a two-stage purification process, namely, a combination of extraction by vapor and vacuum sublimation techniques.

It is known that Th acts as a tetravalent element in most compounds. Thorium tetrachloride (ThCl_4_) has a noticeable volatility, for example, at 650 °C (923 K) its vapor pressure is 10 Torr, which is enough to isolate and sublimate it from GdCl_3_. The temperature dependence of the vapor pressure under solid ThCl_4_ is described by the following equation [16]:lgP (Torr) = −7987/T + 9.57,(2)

ΔH_sub_ = 218 kJ/mole; ΔH_exp_ = 153 kJ/mole.

Uranium exhibits oxidation states +2, +3, +4, +5 and +6 [16] and forms a series of chlorides (Table 5).

The vapor pressure of UCl_4_ is described by Equations (3) and (4) for different temperature ranges [16].
lgP (Torr) = −10,427/T + 13.2995 (T∈623; 742 K),(3)
lgP (Torr) = −7205/T + 9.65 (T∈863; 1063 K)(4)

The partial pressure of UCl_4_ molecules is 70 Torr at 650 °C (923 K). Taking the possibility of reducing the concentration of uranium and thorium during chlorination and subsequent vacuum annealing as the basis, we implemented the following stages of purification of gadolinium chloride.

The chlorination process of the starting gadolinium chloride,Thermal annealing in vacuum.

The chlorination process was carried out using standard laboratory equipment (Figure 4), which included a laboratory quartz reactor, a resistive two-zone furnace with temperature controllers (Termodat-13K6, Perm, Russia), glassy carbon containers and a Drexel vessel.

The initial GdCl_3_ was loaded into a glassy carbon container and placed into the reactor. Chlorination was carried out using high purity hydrogen chloride gas (99.999 vol.%) or NH_4_Cl. The latter disproportionated to HCl and NH_3_ at high temperature. The gas flow was determined by the carrier gas (Ar) and its rate corresponded to 1.0 L/h. The effluent vapors were neutralized in a 10% NaOH water solution with the addition of phenolphthalein as an indicator. As a result of a heterophase reaction of HCl vapor and GdCl_3_, the solid U and Th which had been dissolved in GdCl_3_ crystal lattice could form individual chlorides. We found the conditions (temperature of powder, Ar and HCl flow rates, and process duration) at which U formed the highest chloride, UCl_6_. The partial pressure of UCl_6_ at the reaction temperature of 145–295 °C was several orders higher than that of GdCl_3_ [18]. The argon gas stream picked up the UCl_6_ molecules and carried them away from the GdCl_3_ powder preparation. By this means we succeeded in reducing U content in the GdCl_3_ powder preparation from 2.50 ± 0.10 to 0.10 ± 0.05 ppb.

In the case of thorium impurity, the chlorination process was not as successful when the partial pressure of the ThCl_4_ molecules is compared with those of GdCl_3_. The chlorination only reduced the Th concentration from 10.30 ± 0.50 to 2.85 ± 0.60 ppb.

To reduce Th concentration after the chlorination process was complete we transferred the container with the purified preparation to a laboratory setup. We sublimated the GdCl_3_ powder preparation under high dynamic vacuum (10^−5^ Torr) in a quartz-glass reactor with 50 mm internal diameter and 1000 mm length sealed at one end at which the initial GdCl_3_ powder preparation was placed in a glass-graphite boat. The GdCl_3_ preparation sublimated at heating and flew along the reactor towards the pumping system. Due to the different partial pressure of GdCl_3_, ThCl_4_ and UCl_6_ we separated several fractions. The fraction which was close to the initial preparation had the lowest content of U/Th. The conditions for conducting experiments to reduce the concentrations of thorium and uranium in gadolinium chloride are shown in Table 6.

The best results were obtained using HCl at the annealing temperature of 773 K, followed by vacuum heat treatment at 903 K for 20 h. Samples of gadolinium chloride Gd-05 and Gd-06 reached concentration values for Th and U less than 0.1 ppb and had a total chemical purity of no lower than 99.999 wt% (Figure 5a,b).

### 4.3. Analysis of Commercial Polymers

To create a low-radioactive composite filled with Gd-based preparation one needs a polymer matrix in which the Gd-based preparation would be incorporated. The list of appropriate polymers for usage as construction and supporting materials in the experiments with low-radiation background is comparatively short, namely, polyurethane (PU), polystyrene (PS), polymethylmethacrylate (PMMA) and polyvinyl alcohol (PVA). We analyzed the purity of the above polymers from various manufactures and discovered that the majority of commercial polymers met the requirements for U/Th concentrations for neutron veto. Taking into account the mechanical characteristics, the choice of the material of the organic matrix of the hybrid material was made in favor of polymethyl methacrylate (PMMA), which is the most durable of the materials under consideration at room temperature [19,20,21].

According to the ICP-MS analysis the integral chemical purity of the PMMA preparations from SRI Polymers (Russia) and Voxeljet AG, (Germany) were no lower than 99.996 wt% (Figure 5c,d). The content of uranium and thorium in commercial polymers were relatively low (Figure 6). Additionally, PMMA had the lowest level of U/Th concentrations.

### 4.4. Hybrid Material Fabrication

Protective materials for VETO based on a polymer matrix doped by gadolinium need to have a uniform distribution of gadolinium throughout the composite volume. The fabrication of bulk polymers from PMMA, PS, PU and PVA is a long-term process. In the case of PMMA, the thermal polymerization of a 5 cm thick PMMA sample requires 20–30 days [12]. Obviously, the fabrication of Gd-doped bulk polymer is difficult to realize by Gd oxide impregnation of polymer due to the large difference in densities of gadolinium oxide (ρ = 7.407 g/cm^3^ [10]) and polymers: PMMA (ρ = 1.18 g/cm^3^ [13]), PS (ρ = 1.065–1.125 g/cm^3^ [20]), PU (ρ = 1.1–1.35 g/cm^3^ [21]) and PVA (ρ = 1.19–1.31 g/cm^3^ [21]). Attempts to incorporate a nanosize powder preparation of Gd_2_O_3_ (20–80 nm) did not lead to the required results. Despite the nanoscale nature of the grains, taking into account the large difference in densities, sedimentation of Gd_2_O_3_ occurred. The difference in the gadolinium concentration over the HM thickness exceeded one order of magnitude.

The main idea was that a uniform distribution of Gd can be obtained only if not in suspension, but a true solution would be polymerized. It could be done if a gadolinium coordination compound was dissolved with an organic ligand in methyl methacrylate (MMA) monomer with further polymerization.

Analysis of possible Gd-containing preparations that could dissolve in the starting MMA taking into account that the commercial component showed that the most promising is the preparation of the coordination compound of gadolinium acetylacetonate—Gd(C_5_H_7_O_2_)_3_ (Gd(acac)_3_) [22]. To obtain Gd(acac)_3_ we used an ultra-low-radioactive GdCl_3_. Analysis of commercial preparations of NH_4_OH (25 vol.%) (analytical grade, 23-5, GOST 24147-80), ultra-pure water (see Section 3.1) and acetylacetone C_5_H_8_O_2_, (analytical grade, GOST 10259-78) showed that the contents of U and Th in these preparations were less than 0.1 ppb.

Gd(acac)_3_ was synthesized using the standard procedure [22]. The synthesis details are presented in the Appendix A.

The obtained preparation Gd(acac)_3_ contained a significant amount of water, which negatively affected the polymerization process and the final mechanical characteristics of the Gd-doped PMMA hybrid material. Therefore, the synthesized preparation was subjected to additional annealing in a dynamic vacuum to remove residual water. Gd(acac)_3_ was dried in a dynamic vacuum with exposure at different temperatures. We discovered that the two-step heat treatment allowed not only decreasing of the residual water content but also allowed us to reduce U/Th concentrations to the LD levels (Table 7).

The residual water content in the Gd(acac)_3_ preparations was estimated using IR spectroscopy (FTIR Tensor 27, Bruker, Germany). According to the analysis of IR absorption spectra at 3600–3200 cm^−1^ wavenumbers, attributed to hydroxyl groups [23], we observed a significant decrease of intensity of absorption bands in this region with an increase in the duration of annealing in a dynamic vacuum of 10^−5^ Torr and 423 K (Figure 7). An increase in temperature to 240 °C led to a maximal decrease in the intensity of absorption of OH-groups and a yellowing of the Gd(acac)_3_ sample, which indicated its partial decomposition.

A hybrid material based on PMMA and Gd(acac)_3_ was obtained in two stages. In the first stage, a pre-polymer was prepared as follows. An additive of benzoyl peroxide (initiator of polymerization) was introduced into the MMA monomer. Furthermore, a powder preparation of dehydrated Gd(acac)_3_ and polymer crumb PMMA was added with constant stirring at a temperature of 318 K. In this way, two samples of HM with a nominal gadolinium concentration of 1.0 and 1.5 wt% were prepared.

In the second step, the pre-polymer was placed in a mold, air was removed and sealed. Polymerization was carried out with stepwise heating. At each temperature step, the sample was kept for 10 h. After the polymerization process, Gd-doped HM was additionally annealed at 393 K for 10 h.

To analyze Gd distribution along the thickness of the sample, we cut a thin plate (1 mm) in the vertical direction along the thickness of the sample. The plates were then placed in a scanning electron microscope (VEGA-LMU, Tescan) to assess the Gd-distribution by EDAX technique (Section 3.2) (Figure 8). We observed that in both samples, the relative deviation of Gd concentration from the average value did not exceed 20%, which meets the requirements usually used for DM detectors.

### 4.5. Mechanical Test of Hybrid Material

Mechanical tensile strength tests of the samples showed that at 298 K the characteristics of the HM and PMMA samples practically did not differ (Figure 9). At 77 K, Gd-doped HM was inferior to pure PMMA within the permissible requirements of 20 rel.%. Analysis of the content of uranium and thorium in the HM samples showed that U/Th concentrations were less than LD (U < 0.01 ppb, Th < 0.01 ppb). Thus, the obtained Gd-doped HM meets the requirements usually used for DM detectors.

## 5. Conclusions

As a result of the research, a laboratory technology of new ultra-low radioactive background hybrid material has been developed. The hybrid material is based on a polymer matrix of polymethylmethacrylate (PMMA) in which gadolinium nuclei are homogeneously distributed up to 1.5 wt% concentration in polymer slabs of 5 cm thickness.

A method has been developed for the determination of 65 impurity elements by ICP-MS technique in Gd-based preparations with a determination range of 100–0.01 ppb. The limit of determination was 0.011 ppb for uranium, and 0.016 ppb for thorium.

It was established that the lowest content of U and Th (1.2–0.2 ppb) was discovered in commercial Gd-based preparations manufactured from secondary raw materials (EPA—extraction phosphoric acid) or from mineral raw materials formed in sedimentary rocks (phosphogypsum). The best choice for production of the hybrid material is gadolinium chloride.

The developed two-stage purification process of commercial GdCl_3_ preparations including chlorination annealing and vacuum sublimation procedure makes it possible to reduce U/Th concentrations to the level of 0.06 ± 0.04 for U and 0.05 ± 0.03 for Th.

The purified GdCl_3_ preparations were successfully used for synthesis of low radioactive background Gd coordination compound—acetylacetonate gadolinium (Gd(acac)_3_) with reduced concentration of OH-groups. This preparation was additionally purified to U and Th levels less than 0.011 ppb and 0.016 ppb, respectively. The complete dissolving of Gd(acac)_3_ in methylmethacrylate and further thermal polymerization made it possible to fabricate ultra-low background Gd-doped PMMA blocks which met the requirements usually used for DM detectors.

## Figures and Tables

**Figure 1 materials-14-03757-f001:**
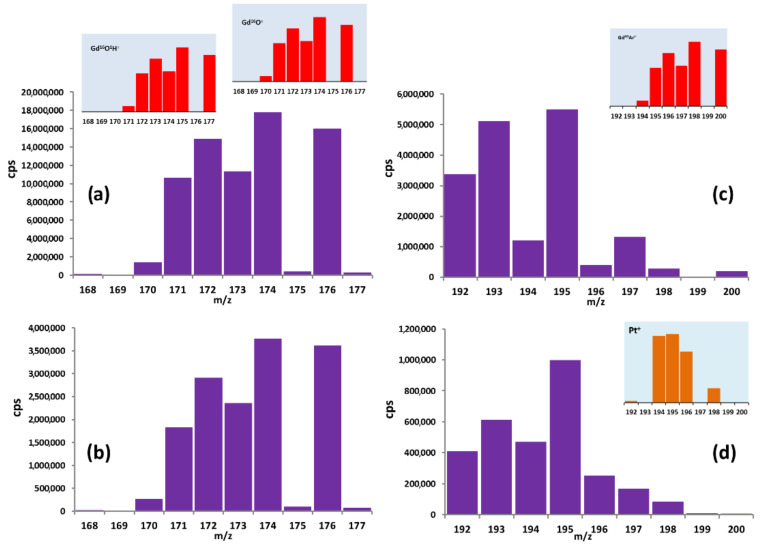
Mass spectra of the sample with a Gd concentration of 0.001 wt% obtained in different modes: (**a**) standard; (**b**) KED for m/z 168–177; (**c**) standard; (**d**) KED for m/z 192–200.

**Figure 2 materials-14-03757-f002:**
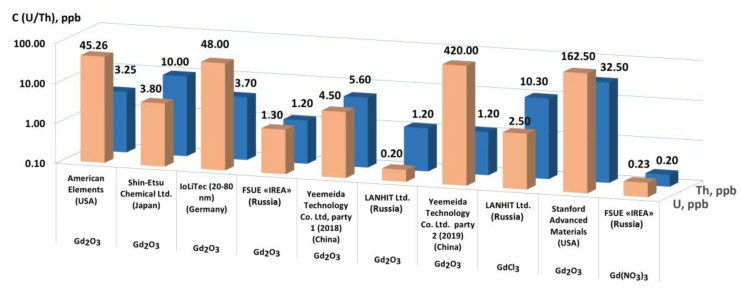
The determined concentration of thorium and uranium in commercial Gd-based preparations.

**Figure 3 materials-14-03757-f003:**
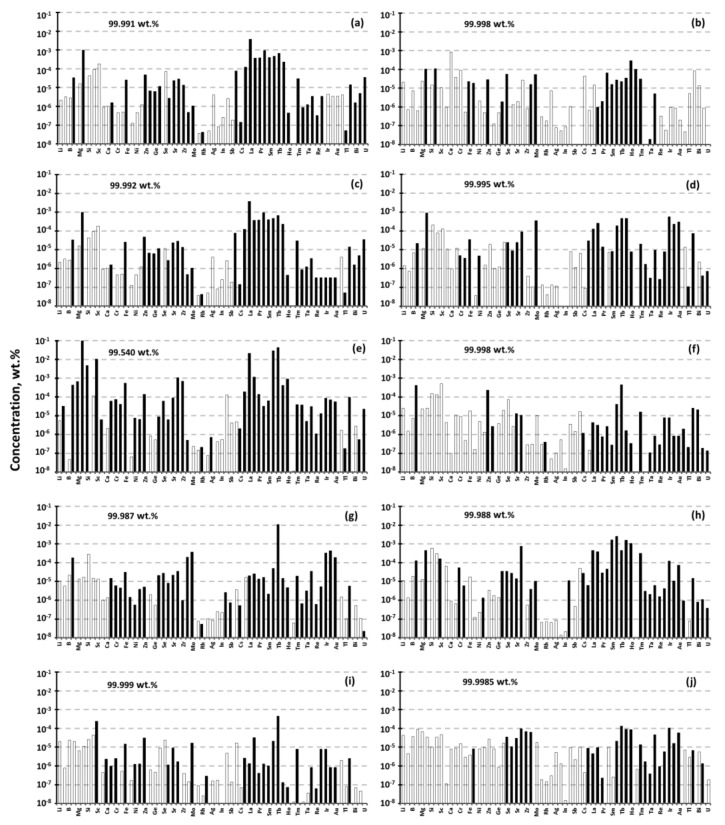
The impurity composition of gadolinium-based preparations of various manufacturers according to the results of ICP-MS analysis: (**a**) Gd_2_O_3_, American Elements; (**b**) Gd_2_O_3_, Party No. 1 (2018) Yeemeida Technology Co. Ltd.; (**c**) Gd_2_O_3_, Stanford Advanced Materials; (**d**) Gd_2_O_3_, Party No. 2 (2019) Yeemeida Technology Co. Ltd.; (**e**) Gd_2_O_3_ IoLiTec (20–80 nm); (**f**) Gd_2_O_3_, FSUE “IREA”; (**g**) Gd(NO_3_)_3_, FSUE “IREA”; (**h**) Gd_2_O_3_, “LANHIT” Ltd.; (**i**) GdCl_3_, “LANHIT” Ltd.; (**j**) Gd_2_O_3_ Shin ETSU (Japan). The empty bars show the limits of determination of the elements.

**Figure 4 materials-14-03757-f004:**
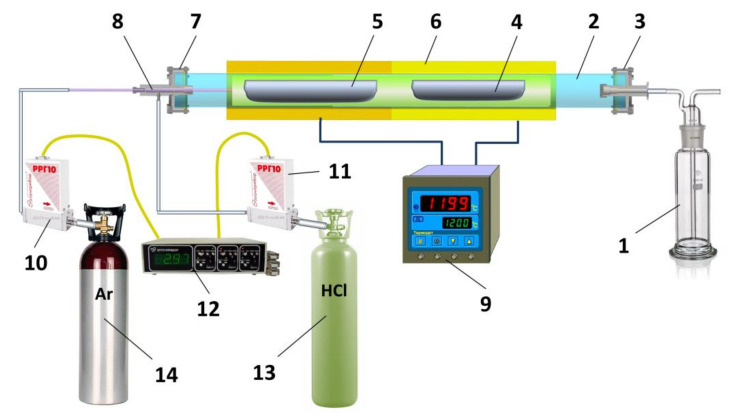
Setup for chlorination of powder preparations: 1—Drexel vessel; 2—quartz-glass reactor valve; 3—outlet pipe system with fungal seal; 4—glassy carbon container for GdCl_3_; 5—glassy carbon container for NH_4_Cl; 6—double zone resistive furnace with cerablanket thermoinsulation; 7—inlet pipe system with fungal seal; 8—capillary tube connector for gas inject; 9—dual channel temperature controller; 10 and 11—mass flow controller; 12—control block for mass flow controllers; 13—HCl cylinder; 14—argon cylinder.

**Figure 5 materials-14-03757-f005:**
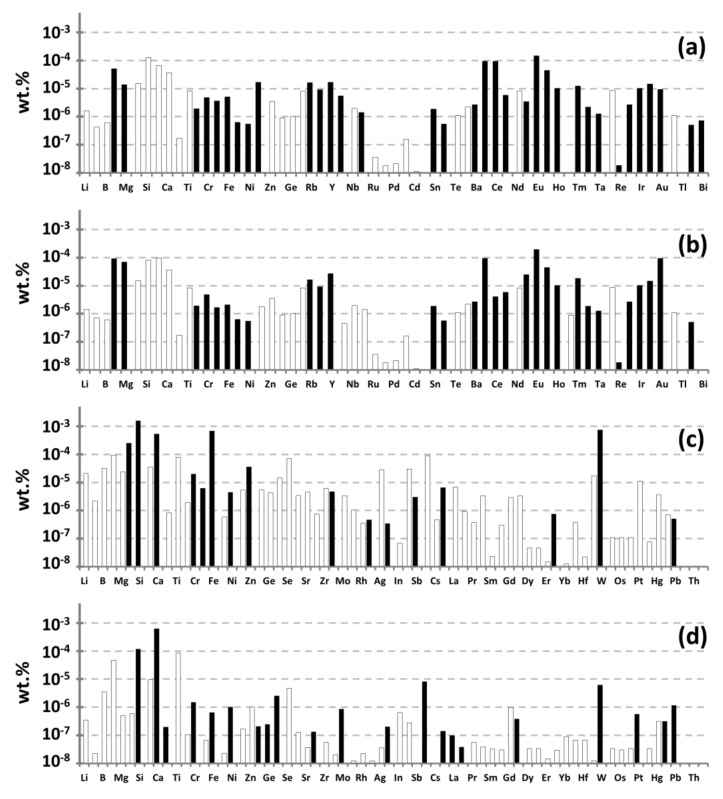
The impurity composition of Gd-05 (**a**), Gd-06 (**b**) GdCl_3_ preparations and PMMA of various manufacturers (**c**) Voxeljet AG, (Germany); (**d**) RND Polymer (Russia) measured by ICP-MS. The empty bars present the limits of determination of the corresponding elements.

**Figure 6 materials-14-03757-f006:**
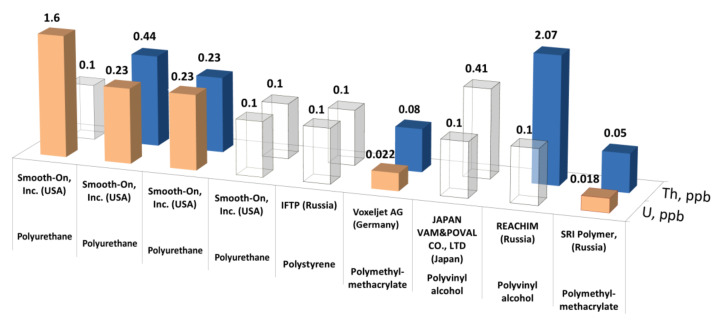
The determined concentration of thorium and uranium in commercial polymers. The empty cubes correspond to the limits of determination.

**Figure 7 materials-14-03757-f007:**
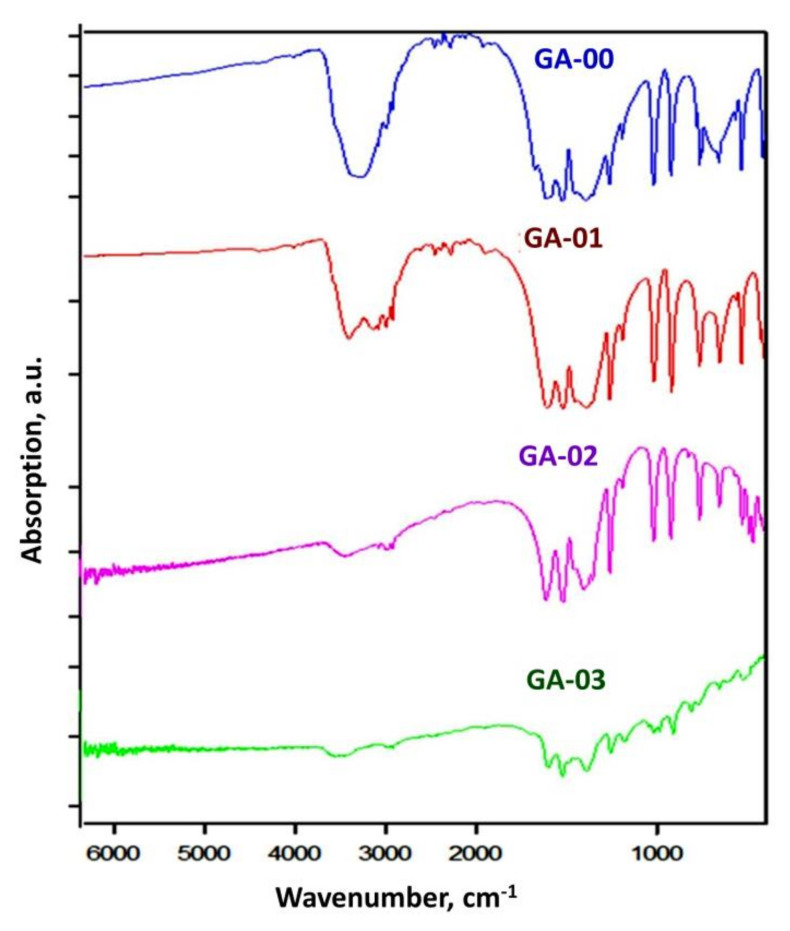
IR absorption spectra of Gd(acac)_3_ preparations after stepwise annealing (ID samples see in Table 7).

**Figure 8 materials-14-03757-f008:**
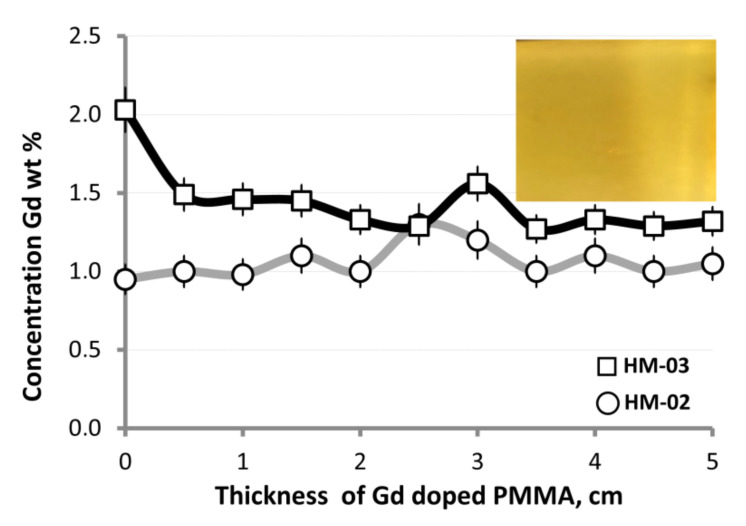
Gd distribution along the HM width determined by EDAX analysis.

**Figure 9 materials-14-03757-f009:**
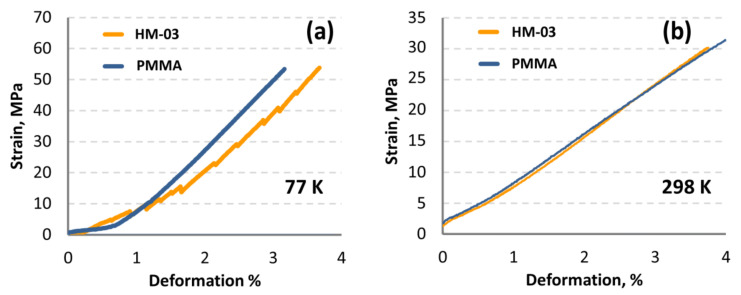
Stress–strain test of nominally pure PMMA and Gd-doped PMMA (HM-03) at room (**a**) and liquid nitrogen (**b**) temperatures.

**Table 1 materials-14-03757-t001:** Comparison of neutron yields * from (α, *n*) reactions for pure polymethylmethacrylate (PMMA), PMMA with gadolinium oxide (1.5 wt% Gd), PMMA with gadolinium acetylacetonate (1.5 wt% Gd) and PMMA with gadolinium fluoride (1.5 wt% Gd).

Material	$Y{(}^{232}\mathrm{Th}),\;10^{-7}\;n/\mathrm{decay}$	$Y{(}^{235}U),\;10^{-7}\;n/\mathrm{decay}$	$Y{(}^{238}U)\;\mathrm{Upper},\;10^{-7}\;n/\mathrm{decay}$	$Y{(}^{238}U)\;\mathrm{Middle},\;10^{-7}\;n/\mathrm{decay}$	$Y{(}^{238}U)\;\mathrm{Lower},\;10^{-7}\;n/\mathrm{decay}$
PMMA	13.18	14.02	2.11	8.47	1.16
PMMA + Gd_2_O_3_ (1.5 wt% Gd)	13.10	13.94	2.09	8.42	1.15
PMMA + Gd(C_5_H_7_O_2_)_3_ (1.5 wt% Gd)	13.12	13.96	2.10	8.44	1.15
PMMA + GdF_3_ (1.5 wt% Gd, ≈ 0.5 wt% F)	18.41	19.41	2.50	11.74	1.51

* The neutron yields are given per single decay of the parent nucleus in a particular chain

**Table 2 materials-14-03757-t002:** The operating mode of the NexION 300D instrument for conducting impurity analysis of samples with Gd matrix element.

Nebulizer type	Concentric (Meinhard), PFA
Spray chamber	Scott double-pass chamber, PFA
Argon flow rate, L/min	
through the nebulizer	0.96
Plasma-forming	15
Auxiliary	1.2
Generator power, W	1450
Collision gas (He) flow rate, L/min	4.6
Number of scan cycles	8

**Table 3 materials-14-03757-t003:** Possible isobar overlays in the analysis of a “blank” solution obtained after dissolution and analyzed by inductively coupled plasma mass-spectrometry (ICP-MS).

Element	Isotope (Natural Abundance, %) [13]	Interferences	Element	Isotope (Natural Abundance, %) [13]	Interferences
Mg	24 (78.99)	^12^C^12^C^+^	Fe	56 (91.72)	^40^Ar^16^O^+^
Al	27 (100)	^12^C^15^N^+^ ^12^C^14^N^1^H^+^	Co	59 (100)	^40^Ar^18^O^1^H^+^
K	39 (93.25)	^38^Ar^1^H^+^	Ni	60 (26.23)	^40^Ar^18^O^1^H^1^H^+^
Ca	40 (96.941)	^40^Ar^+^	V	51 (99.750)	^38^Ar^12^C^1^H^+^
Mn	55 (100)	^40^Ar^14^N^1^H^+^	Cr	52 (83.789)	^40^Ar^12^C^+^
Se	76 (9.36) 77 (7.63) 78 (23.78) 80 (49.61)	^152^Gd^++^ ^154^Gd^++^ ^156^Gd^++^ ^160^Gd^++^	Yb	171 (14.3) 172 (21.3) 173 (16.12) 174 (31.8)	^155^Gd^16^O^+ 154^Gd^16^O^1^H^+^ ^156^Gd^16^O^+ 155^Gd^16^O^1^H^+^ ^157^Gd^16^O^+ 156^Gd^16^O^1^H^+^ ^158^Gd^16^O^+ 157^Gd^16^O^1^H^+^
Dy	156 (0.06) 158 (0.10) 160 (2.34) 161 (18.91)	^156^Gd^+^ ^158^Gd^+^ ^160^Gd^+^ ^160^Gd^1^H^+^	Tb	159 (100)	^155^Gd^16^O^+ 154^Gd^16^O^1^H^+^ ^156^Gd^16^O^+ 155^Gd^16^O^1^H^+^ ^157^Gd^16^O^+ 156^Gd^16^O^1^H^+^ ^158^Gd^16^O^+ 157^Gd^16^O^1^H^+^
Sm	152 (26.75) 154 (22.75)	^152^Gd^+^ ^154^Gd^+^	Tm	169 (100)	^158^Gd^1^H^+^
Hf	174 (0.16) 176 (5.26) 178 (27.28)	^158^Gd^16^O^+^ ^160^Gd^16^O^+^ ^162^Gd^16^O^+^	Lu	175 (97.41) 176 (2.59)	^159^Gd^16^O^+^ ^160^Gd^16^O^+^

**Table 4 materials-14-03757-t004:** Recommended isotopes for adjusting difficult-to-detect elements in the analysis of gadolinium-based solutions.

Isotope	Natural Abundance, %	Isotope	Natural Abundance, %
^82^Se	82	^178^Hf	178
^147^Sm	147	^188^Os	188
^151^Eu	151	^194^Pt	194
^163^Dy	163		

**Table 5 materials-14-03757-t005:** The physical properties of thorium and uranium chlorides [16,17,18].

Compound	ΔH^0^_298_, kJ/mole	T_m,_ K	T_boil_._,_ K
ThCl_4_	−1189	1043	1193
UCl_3_	−891,2	1108	1657
UCl_4_	−1051	863	1071
UCl_5_	−1094	Unstable disproportionate	-
UCl_6_	−1133	450.5	823

**Table 6 materials-14-03757-t006:** The determined concentration of thorium and uranium in GdCl_3_ samples synthesized at different conditions.

Sample ID	Step	T, K	Time, h	Atmosphere	Concentration, ppb	
					Th	U
Gd-01	mix (NH_4_Cl:GdCl_3_ = 10:1)	593	3	HCl, NH_3_	2.45 ± 0.08	0.58 ± 0.01
	Vacuum annealing	873	6	10^−5^ Torr		
Gd-02	Atmosphere NH_4_Cl	613	5	HCl, NH_3_	3.23 ± 0.12	1.21 ± 0.09
	Vacuum annealing	873	6	10^−5^ Torr		
Gd-03	Chlorination	423	3	HCl	5.26 ± 0.35	1.09 ± 0.02
	Vacuum annealing	873	6	10^−5^ Torr		
Gd-04	Chlorination	573	5	HCl	4.30 ± 0.09	0.30 ± 0.01
	Vacuum annealing	873	9	10^−5^ Torr		
Gd-05	Chlorination	773	6	HCl	0.08 ± 0.01	0.09 ± 0.01
	Vacuum annealing	903	12	10^−5^ Torr		
Gd-06	Chlorination	773	9	HCl	0.07 ± 0.01	0.06 ± 0.01
	Vacuum annealing	903	20	10^−5^ Torr		

**Table 7 materials-14-03757-t007:** Experimental conditions for the stepwise annealing of Gd(acac)_3_ preparations.

ID Sample	T K	Time, h	Annealing Condition	C_Th_, ppb	C_U_, ppb
GA-00	353	10	air	0.14 ± 0.02	1.60 ± 0.03
GA-01	353	5	10^−5^ Torr (dynamic)	0.08 ± 0.01	0.40 ± 0.01
GA-02	353→423	1→3	10^−5^ Torr (dynamic)	<0.016	<0.011
GA-03	353→423→513	1→3→3	10^−5^ Torr (dynamic)	<0.016	<0.011

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
