# Peer review of "Hybrid Ultra-Low-Radioactive Material for Protecting Dark Matter Detector from Background Neutrons"

_materials, 2021, doi:10.3390/ma14133757_

Round 1

Reviewer 1 Report

Report on paper: Materials-1243878

As stated in the title a new material is proposed, with the definite aim of shielding massive DM experiments from unwanted neutron background events. The objective of the work is reasonably well stated and the experimental work carried out in a competent way. Apart from some obvious typos and minor corrections which are listed at the end of this report, there a couple of points (mentioned in the introduction) which deserve to be better clarified prior to publication.

In the introduction, reference is made to measurement of DM particles; I understand that the Authors refer to detection of nuclear recoil events, but this cannot be assumed a priori. Also, this may be not clear to the general reader; I think the Authors should spend a few words in order to clarify this concept.
On page 2, the Authors underline the importance of the gamma events (coming from thermal neutron absorption in Gd) in acting as signature events of neutron absorption. But, looking from a different angle, one must also consider that in very low background experiments these events can cause background and even mimic events of unknown nature; for instance, there may be spillover to the low-energy response of the detector with pollution in the nuclear recoil energy region.  Furthermore, the gamma energy range involved (up to 8 MeV) is not easily amenable to calibration in underground laboratory conditions; in these conditions, typical energies which can be explored with lab sources are limited to about 2 MeV. Has this problem been studied in detail ? Do the Authors have some numerical estimates of the possible effects ? Since these experiments are scheduled to run for many years, a proper understanding of all potential background events is mandatory; in order to fully appreciate the impact of this work, I think that the Authors should expand this point.  

Minor: 
Here and there in the text: Activity should read Bq/Kg and not Bk/Kg
Table 6: "Vacuum" instead of "Vacuun"
Formulas 2,3,4: lg is the natural log ? Which units for T ? Also, I have a professional habit of writing math functions with non-dimensional arguments, so I find this writing style a bit confusing. However, if this is standard practice in the field, we can live with it.  

Author Response

Comments and Suggestions for Authors

Report on paper: Materials-1243878

As stated in the title a new material is proposed, with the definite aim of shielding massive DM experiments from unwanted neutron background events. The objective of the work is reasonably well stated and the experimental work carried out in a competent way. Apart from some obvious typos and minor corrections which are listed at the end of this report, there a couple of points (mentioned in the introduction) which deserve to be better clarified prior to publication.

In the introduction, reference is made to measurement of DM particles; I understand that the Authors refer to detection of nuclear recoil events, but this cannot be assumed a priori. Also, this may be not clear to the general reader; I think the Authors should spend a few words in order to clarify this concept.

We agree that a clarification on the DM detection method(s) is required in the introduction. The first few sentences have been replaced with the following text:

The direct search for dark matter (DM) particles is one of the most important and fundamental problems in modern physics. The particles could be detected through their elastic scattering on target nuclei. Low-energy (<100 keV) nuclear recoils are produced as a result of this process and they are registered by the experimental setup. This is a main approach for direct searching for dark matter particles. In this case, the neutron background is one of the most unpleasant and requires maximum suppression. The interaction of neutron with a target leads to a nuclear recoil event and, therefore, may mimic the scattering of DM particles.

On page 2, the Authors underline the importance of the gamma events (coming from thermal neutron absorption in Gd) in acting as signature events of neutron absorption. But, looking from a different angle, one must also consider that in very low background experiments these events can cause background and even mimic events of unknown nature; for instance, there may be spillover to the low-energy response of the detector with pollution in the nuclear recoil energy region.  

When gamma interacts with target of the DM detector it produces an electron recoil. The pulse shapes of the primary scintillation signals (S1) are significantly different for the nuclear and electron recoils, so that gamma background can be suppressed by applying a pulse shape discrimination. Moreover, it’s expected the neutron absorption will mostly take place in a special part of the DM detector called veto. This part is usually equipped with PMTs or SiPMs and gammas are detected there. We think the proposed hybrid material will be used in the detector veto. Last but not least, gammas have pretty high energies in case of neutron capture on Gd, and therefore these gammas can be detected effectively.

Furthermore, the gamma energy range involved (up to 8 MeV) is not easily amenable to calibration in underground laboratory conditions; in these conditions, typical energies which can be explored with lab sources are limited to about 2 MeV. Has this problem been studied in detail? Do the Authors have some numerical estimates of the possible effects? Since these experiments are scheduled to run for many years, a proper understanding of all potential background events is mandatory; in order to fully appreciate the impact of this work, I think that the Authors should expand this point.  

Gadolinium compounds are widely used for neutron capture and subsequent identification of relevant events by the delayed coincidence method in neutrino physics (Daya Bay, RENO, TAO (the JUNO near detector), Double Chooz experiments). This approach was borrowed into dark matter physics from there. The gamma spectrum resulting from neutron capture on gadolinium is well known and studied. It can be obtained in the detector by using a neutron calibration source (241Am-13C, 241Am-9Be, D-D guns). Since the resulting spectrum is quite complex, Monte Carlo simulations are used to interpret it and calibrate the energy scale.

In other words, we rely on standard experimental methods in our study. And it means there is no need to study any calibration-related points including the feasibility of using high-energy gammas.   

Minor: 
Here and there in the text: Activity should read Bq/Kg and not Bk/Kg
Table 6: "Vacuum" instead of "Vacuun"

Both issues were fixed

Formulas 2,3,4: lg is the natural log ?

lg is log10 logarithm which is commonly used for technical applications.

Which units for T ?

We corrected this point. All temperatures in the article are now presented in K (Kelvin).

Also, I have a professional habit of writing math functions with non-dimensional arguments, so I find this writing style a bit confusing. However, if this is standard practice in the field, we can live with it.

You are right.

The correct expression for the pressure looks like that

Lg(P/P0)= -A/kT+B/k….

Here k is Botlzman constant,  P is the partial pressure of a substance and P0 is the pressure at a standard state. In chemical thermodynamics the common standard state is determined at P0=1 atm. And the majority of reference data in thermodynamics books are associated with this standard state. On the other hand, there are many examples when the partial pressure is measured in Torr for the practical usage. So, in this case the B-coefficient is transformed in the equation for the appropriate unit. In the present submission, we used the literature data for the equations for the partial pressure using the original units.

Reviewer 2 Report

I have reviewed the manuscript "Hybrid ultra low-radioactive material for protecting dark matter detector from background neutrons" by Zykova et al. This paper describes the development of Gd doped polymers that would be useful as active or passive shields of cosmogenic neutrons in low background counting experiments and other applications. Overall the scientific work described is sound and it is described in good detail. This paper should be of interest to researchers working in this field. My main criticism is that the English grammar and usage needs improvement throughout. I recommend the authors enlist a native English speaking colleague to assist with that. I also have a few specific comments, below. After revision this manuscript will be suitable for publication.

1. I don't understand Eq. 1. The units seem to be mass on the right and dimensionless on the left. Also, what is epsilon? I don't understand what "efficiency of neutron selection" means.

2. Table 1: Why are the U-238 neutron yields separated into three groups? Isn't the total of these the most relevant and important?

3. lines 55, 113, and elsewhere: uBk should be typeset using the Greek prefix "mu"

4. All acronyms should be defined at first use, such as ED (line 64), RE (line 258), VETO (line 343), and elsewhere.

Author Response

Comments and Suggestions for Authors

I have reviewed the manuscript "Hybrid ultra low-radioactive material for protecting dark matter detector from background neutrons" by Zykova et al. This paper describes the development of Gd doped polymers that would be useful as active or passive shields of cosmogenic neutrons in low background counting experiments and other applications. Overall the scientific work described is sound and it is described in good detail. This paper should be of interest to researchers working in this field. My main criticism is that the English grammar and usage needs improvement throughout. I recommend the authors enlist a native English speaking colleague to assist with that. I also have a few specific comments, below. After revision this manuscript will be suitable for publication.

  1. I don't understand Eq. 1. The units seem to be mass on the right and dimensionless on the left. Also, what is epsilon? I don't understand what "efficiency of neutron selection" means.

Ai is the total specific activity (per unit mass of the material) for a part or complete decay chain designated by the index i . It wasn’t stated clearly, so we agree the reader might have thought that the dimensions do not match in Equation 1.

Corrected.

ε is the detector efficiency of neutron selection.

We hope that the meaning is clear now.

  1. Table 1: Why are the U-238 neutron yields separated into three groups? Isn't the total of these the most relevant and important?

We added the following comments in the text.

It’s done this way because the secular equilibrium might be broken for the 238U decay chain, but it is expected to be preserved in each sub-chain [10]. Unlike other isotopes in the radioactive series 226Ra and 210Pb have long half-lifes, 1600 and 22 years, respectively, and can be accumulated in materials due to different processes [10].

The respective reference is

[10] Westerdale, S.; Meyers, P. D. Radiogenic Neutron Yield Calculations for Low-Background Experiments. Nucl. Instruments Methods Phys. Res. Sect. A Accel. Spectrometers, Detect. Assoc. Equip. 2017, 875, 57–64. https://doi.org/10.1016/j.nima.2017.09.007.

  1. lines 55, 113, and elsewhere: uBk should be typeset using the Greek prefix "mu"

 Corrected   -  μBq/kg

  1. All acronyms should be defined at first use, such as ED (line 64), RE (line 258), VETO (line 343), and elsewhere.

Corrected